# Evaluation of the Jichi Medical University diverticular hemorrhage score in the clinical management of acute diverticular bleeding with emergency or elective endoscopy: A pilot study

Takeshi Uehara[1]*, Satohiro Matsumoto[1], Hiroyuki Tamura[2], Masahiro Kashiura[2], Takashi Moriya[2], Kenichi Yamanaka[3], Hakuei Shinhata[3], Masanari Sekine[1], Hiroyuki Miyatani[1], Hirosato Mashima[1]

1 Department of Gastroenterology, Jichi Medical University Saitama Medical Center, Saitama, Japan,
2 Department of Emergency Medicine, Jichi Medical University Saitama Medical Center, Saitama, Japan,
3 Department of Gastroenterology, Saitama Citizens Medical Center, Saitama, Japan

* t-uehara@jichi.ac.jp

**Data Availability Statement:** All relevant data are within the manuscript.

## Abstract

### Background and aims

Emergency endoscopic hemostasis for colonic diverticular bleeding is effective in preventing serious consequences. However, the low identification rate of the bleeding source makes the procedure burdensome for both patients and providers. We aimed to establish an efficient and safe emergency endoscopy system.

### Methods

We prospectively evaluated the usefulness of a scoring system (Jichi Medical University diverticular hemorrhage score: JD score) based on our experiences with past cases. The JD score was determined using four criteria: CT evidence of contrast agent extravasation, 3 points; oral anticoagulant (any type) use, 2 points; C-reactive protein ≥1 mg/dL, 1 point; and comorbidity index ≥3, 1 point. Based on the JD score, patients with acute diverticular bleeding who underwent emergency or elective endoscopy were grouped into JD ≥3 or JD <3 groups, respectively. The primary and secondary endpoints were the bleeding source identification rate and clinical outcomes.

### Results

The JD ≥3 and JD <3 groups included 35 and 47 patients, respectively. The rate of bleeding source identification, followed by the hemostatic procedure, was significantly higher in the JD ≥3 group than in the JD <3 group (77% vs. 23%, p <0.001), with a higher JD score associated with a higher bleeding source identification rate. No significant difference was observed between the groups in terms of clinical outcomes, except for a higher incidence of rebleeding at one-month post-discharge and a higher number of patients requiring

**Funding:** The authors received no specific funding for this study.

**Competing interests:** The authors have declared that no competing interests exist.

interventional radiology in the JD ≥3 group than in the JD <3 group. Subgroup analysis showed that successful identification of the bleeding source and hemostasis contributed to a shorter hospital stay.

## Conclusion

We established a safe and efficient endoscopic scoring system for treating colonic diverticular bleeding. The higher the JD score, the higher the bleeding source identification, leading to a successful hemostatic procedure. Elective endoscopy was possible in the JD <3 group when vital signs were stable.

## Introduction

Colonic diverticulosis has historically been prevalent among Westerners. However, in recent years, due in part to the aging population and the westernization of diet, it has also been on the rise in Japan [1–3]. Along with this trend, the prevalence of diverticular-related diseases is also increasing [2, 4]. In particular, colonic diverticular hemorrhage accounts for approximately 23% of all diverticular-related diseases [4] and is a leading cause of lower gastrointestinal (GI) bleeding, accounting for approximately 30–50% of cases [4–7].

Guidelines from the Japanese Gastroenterological Association [8], European Society of Gastrointestinal Endoscopy [9], and American Gastroenterological Association [10] recommend emergency endoscopy within 24 hours of presentation of lower GI bleeding. However, the clinical rationale for this recommendation is not very clear, and the level of evidence is low [8–10]. Compared to other forms of lower GI bleeding, diverticular bleeding is associated with a higher risk of serious consequences, such as the need for blood transfusion, interventional radiology (IVR), or surgical hemostasis [11, 12]. It is, therefore, important to investigate how to best manage colonic diverticular bleeding to avoid such consequences, where possible.

At our hospital, we have been performing an aggressive endoscopic intervention for lower GI bleeding with the belief that aggressive endoscopy positively impacts patient outcomes. The advantages of emergency endoscopy are that the cause of bleeding can be identified earlier, the presence or absence of active bleeding can be directly determined, and the identification of the bleeding source can be followed immediately by hemostatic treatment [13]. We believe that these advantages will help shorten hospital stays, avoid emergency surgeries and blood transfusions, optimize patients' systemic management, and reduce medical costs [13–16]. However, there are many reports questioning the therapeutic benefits of emergency endoscopy owing to the low identification rate of the bleeding source, the fact that the majority of diverticular bleeding cases have spontaneous hemostasis [17–20], and the high incidence of rebleeding after hemostatic treatment [14, 21–23]. The fact that emergency colonoscopy is a burdensome procedure for both patients and providers and the accumulation of knowledge about diverticular bleeding, prompted us to review our treatment strategy.

Factors affecting the bleeding source identification rate include those related to the operability of the endoscope and endoscopic techniques, such as the possible implementation of endoscopy within 24 hours of the patient's arrival, endoscopist's skill, the use of a water jet scope or endoscope tip hood, and the use of oral rinsing agents [14, 24–26]. Several studies have reported that the development of diverticular hemorrhage is affected by: local factors, such as diverticulitis; systemic factors, such as hypertension, atherosclerosis-related diseases, and increased organ fat [27, 28]; and other factors, such as alcohol consumption, smoking

index ≥400, nonsteroidal anti-inflammatory drugs (NSAIDs), low-dose aspirin, and non-aspirin antiplatelet drugs [29–31]. However, there are no studies on factors that may influence the ease of identifying the source of bleeding. When the bleeding site is identified, the hemostatic procedure can be performed immediately, which may improve the clinical course of patients [13, 32].

We previously performed a retrospective analysis of 178 consecutive patients with colonic diverticular bleeding treated at our hospital over 5 years, from 2010 to 2014, dividing the patients into two groups: those in whom the bleeding source could be identified by emergency endoscopy and those in whom it could not. No study has examined whether emergency endoscopy should be performed for colonic diverticular bleeding by using a scoring system. As such, we developed a scoring system called the Jichi Medical University diverticular hemorrhage score (JD score). We hypothesized that using the JD score to pare down the number of patients undergoing emergency endoscopy would allow us to perform effective management of diverticular hemorrhage without relying on the judgment of a gastroenterologist, thereby contributing to a reduction in the burden on both patients and providers.

This study aimed to determine the usefulness of the JD score in the clinical management of acute diverticular bleeding using emergency or elective endoscopy.

## Materials and methods

### Patient selection (inclusion and exclusion criteria)

In this prospective multicenter study, patients who visited Jichi Medical University Saitama Medical Center or Saitama Citizens Medical Center between May 2018 and March 2020 with a chief complaint of bloody stools were assessed for eligibility. Jichi Medical University Saitama Medical Center is a tertiary medical institution with 20 endoscopists, and Saitama Citizens Medical Center is a secondary medical institution with four endoscopists.

The inclusion criteria comprised patients with suspected colonic diverticular hemorrhage. Patients were excluded if they: 1) were <20 years of age and neither the patient nor a representative gave consent, 2) were suspected of having any type of bleeding other than diverticular bleeding, 3) could not undergo a contrast-enhanced computed tomography (CE-CT) scan due to renal dysfunction or allergy to contrast agent, 4) could not undergo endoscopy at the time of group allocation for this study because of the decision of the admitting physician, or 5) had an episode(s) of bloody stool within 30 days before presentation, because it may suggest rebleeding.

This study was approved by the Ethics Committee of Jichi Medical University Saitama Medical Center (Rin S17-034) and the Ethics Committee of Saitama Citizens Medical Center (2018–18).

### The JD score

In our previous report, we developed a scoring system (JD score), where 3 points were assigned for CT evidence of contrast agent extravasation, 2 points for oral anticoagulant (any type) use, and 1 point each for C-reactive protein (CRP) ≥1 mg/dL and comorbidity index [33] ≥6. The JD score cut-off of 3 predicted the identification of the bleeding source with 80% sensitivity and 81% specificity [34].

### Allocation of patients according to the JD score

When a patient with lower GI bleeding was transported to or presented to the emergency department, an emergency department physician performed the initial examination, including

CE-CT and treatment. Patients were assessed for eligibility to participate in this study. After excluding patients who met the exclusion criteria, the remaining patients who provided informed consent were classified according to their JD scores. For patients with a JD score $\geq 3$, a physician on duty in the gastroenterology department was contacted, and an emergency endoscopy was performed as soon as possible under systemic clinical management. For patients with a JD score $<3$, systemic management was initially applied, and endoscopy was performed during weekday working hours. In cases where the attending physician made a decision that deviated from the protocol from the viewpoint of priority to the patient and safety assurance or based on the patient's clinical course, that decision was given priority, and the patient was excluded from the analysis. Patients, who were shown to have bleeding other than diverticular hemorrhage by endoscopy, were also excluded. Finally, patients with acute diverticular bleeding who underwent emergency or elective endoscopy according to the JD score were grouped into either JD $\geq 3$ or JD $<3$ groups, respectively.

## Patients' medical care

Similar systemic management procedures such as fluid replacement and blood transfusion were performed in both groups. When the patient's hemoglobin level was likely to fall below 7 g/dL, a blood transfusion was performed in sufficient volume to maintain the hemoglobin level at 7–9 g/dL [35, 36].

Bowel preparation with polyethylene glycol (PEG) solution was performed on the day of endoscopy, if possible. In patients with a JD score $\geq 3$, bowel preparation was rarely performed given the constraints of the 5-hour window for early endoscopy. Bowel preparation was not performed if the patient was expected to have few residual stools based on the duration of fasting and the frequency of defecation after admission. When bowel preparation was not performed, water-jet bowel lavage with PEG solution was often used as an alternative [37].

In addition to the identification of active bleeding, the definition of bleeding source identification included the identification of stigmata of recent hemorrhage such as non-bleeding visible vessels and adherent clots [13]. When the source of bleeding was identified endoscopically, endoscopic hemostatic procedure was applied, including clipping and endoscopic band ligation (EBL), which was selected at the discretion of the endoscopist. If hemostasis was not achieved endoscopically, IVR or surgery was considered. Rebleeding was defined as fresh bloody stools after endoscopy, which confirmed hemostasis, or bloody stools with a decrease of more than 2 g/dL of Hb. The patient was allowed to eat once hemostasis was confirmed. The physician in charge determined whether to withdraw or resume antiplatelet or anticoagulant medication, depending on each patient's status.

## End points

A hemostatic procedure was performed in all cases in which the bleeding source was identified. Hemostatic procedures may improve the clinical course for patients [13, 32]. We hypothesized that in patients with a high probability of identifying the bleeding source, emergency endoscopy could be undertaken, while elective endoscopy could be undertaken in patients with a low probability of identifying the bleeding source when vital signs were stable. In diverticular hemorrhage, identification of the bleeding source is key to confirming the diagnosis and achieving hemostasis. Therefore, the bleeding source identification rate was set as the primary endpoint.

The secondary endpoints were the length of hospital stay, percentage of patients requiring transfusion, transfusion volume, rebleeding rate, and adverse events. Each was subjected to a comparative analysis between the two groups.

Furthermore, to evaluate the benefit of hemostatic treatment, the patients were divided into those in whom the bleeding source could be identified (identified group) and those in whom it could not (unidentified group) and subjected to a subgroup analysis.

## Statistical analysis

The rationale for setting the target sample size was as follows. In our previous study, the bleeding source identification rate in patients with bleeding colonic diverticula who underwent emergency lower GI endoscopy was 26%. In contrast, the application of our scoring system resulted in a 67% bleeding source identification rate in the group of patients with a JD score ≥3 [34]. Based on these known data, a paired *t*-test was used to determine the sample size needed to detect a significant difference in the emergency endoscopy group with 80% power and 5% alpha error, which we calculated to be 27 patients. In this study, patients with a JD score ≥3 were assigned to receive emergency endoscopy and patients with a JD score <3 were assigned to receive elective endoscopy. Given that the ratio of patients with a score ≥3 to those with a score <3 in the previous study was 1:2, the total required sample size was calculated to be 81. Considering the possibility of dropouts and the inclusion of patients with bleeding types other than diverticular bleeding, the final minimum sample size required for the study to be feasible was set at 90 patients.

Data are expressed as median (min-max) or percentage. Statistical analysis was performed using Mann-Whitney U (non-parametric test) and chi-square tests. All statistical analyses were performed using EZR (Jichi Medical University Saitama Medical Center, Saitama, Japan), a graphical user interface for R (The R Foundation for Statistical Computing version 2.13.0) [38]. Differences were considered significant when p-values were <0.05.

## Results

### The JD ≥3 and JD <3 groups (Fig 1)

During the study period, 126 patients presented with lower GI bleeding. Of these, 29 patients were suspected to have a cause of bleeding other than colonic diverticular bleeding, based on medical history and CT findings. Four patients could not undergo CE-CT because of renal dysfunction or other reasons. Two patients refused to participate in the clinical trial. The remaining 91 patients with suspected colonic diverticular bleeding were included in the study. Of these, 39 had a JD score ≥3 and 52 had a JD score <3. Of the 39 patients with a JD score

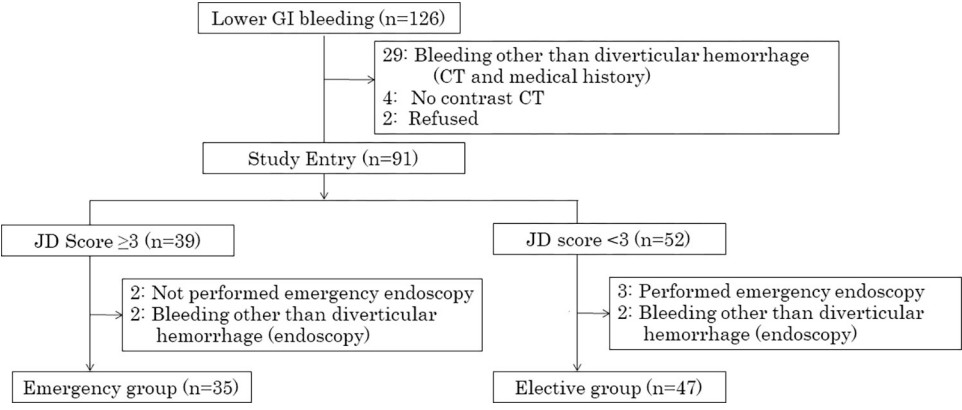

**Fig 1. Patient eligibility assessment, classification by JD score, and exclusion.**

≥3, two patients did not undergo emergency endoscopy and were treated conservatively at the physician's discretion because they were too old to consent to emergency endoscopy or their vitals were unstable. Two patients were judged unlikely to have colonic diverticular bleeding based on subsequent endoscopy. Of the 52 patients with a JD score <3, three underwent emergency endoscopy on holidays or during the night at the discretion of their physicians due to a large amount of blood in the stool or unstable vital signs. Two patients were judged unlikely to have colonic diverticular bleeding based on subsequent endoscopy.

After excluding these four patients (10%, 4/39) with a JD score ≥3 and five patients (10%, 5/52) with a JD score <3, 35 patients in the JD ≥3 group and 47 in the JD <3 group were included in the analysis.

## Background patient characteristics

No significant differences were noted in terms of patient background factors such as age, sex, and underlying diseases between the two groups (Table 1). The number of bloody stool episodes before the hospital visit was significantly lower in the JD ≥3 group.

Among the JD score components, the incidence of CT evidence of intestinal leakage of contrast agent was significantly higher in the JD ≥3 group (91%) than in the JD <3 group (0%; p <0.001). In contrast, the two groups showed no significant differences in CRP levels, comorbidity index, or anticoagulant use.

## Clinical outcomes

The clinical course of the patients is shown in Table 2. The bleeding source identification rate at initial endoscopy was significantly higher in the JD ≥3 group (77%) than in the JD <3 group (23%) (p <0.001). As such, higher JD scores were commensurate with a higher bleeding source identification rate (p <0.001; Fig 2). Endoscopic hemostasis with clips was performed in all cases in which the source of bleeding could be identified. The hemostasis success rate in patients who underwent endoscopic procedures was 96% (26/27) in the JD ≥3 group and 100% (11/11) in the JD <3 group. One patient in the JD ≥3 group who failed to achieve hemostasis with hemoclips was successfully treated with additional IVR. Regarding the time of the procedure, the time from arrival at the hospital to the start of endoscopy was 2.2 (1.2–8.0) hours in the JD ≥3 group and 18.2 (1.8–77.0) hours in the JD <3 group (p <0.01). The endoscopic procedure time was significantly longer in the JD ≥3 group, 70 (25–169) minutes than in the JD <3 group, 45 (19–112) minutes (p <0.01). The percentage of patients who received pretreatment/laxatives before endoscopy was 5% in the JD ≥3 group and 85% in the JD <3 group, indicating that pretreatment was rarely performed in the JD ≥3 group (p <0.01). The percentage of patients who received blood transfusion and the transfusion volume were not significantly different. The number of critical patients requiring >10 units of transfusion was higher in the JD ≥3 group than in the JD <3 group, but the difference was not significant. Among them, two patients in the JD ≥3 group were transfused with >20 units of blood (36 and 22 units, respectively). In the patient who required 36 units, the bleeding source could not be identified on the initial endoscopy. Rebleeding occurred repeatedly, finally necessitating hemostatic treatment with IVR. This patient spent 15 days in the hospital, including 7 days in the intensive care unit (ICU), and was the only patient in this study admitted to the ICU. In the patient who required 22 units of blood, the bleeding source was identified on initial endoscopy, and endoscopic clipping was successfully performed. However, rebleeding continued in this patient, and hemostasis with IVR was required. In total, five patients underwent IVR during hospitalization, all of whom were in the JD ≥3 group (p = 0.027). However, there was one patient in the JD <3 group who was hospitalized for 17 days. This patient had diverticulitis in

**Table 1. Baseline characteristics.**

| | JD ≥3 group (n = 35) | JD <3 group (n = 47) | p-value |
|---|---|---|---|
| Age | 73 (22–94) | 76 (40–91) | 0.343 |
| Male number | 25 (71%) | 35 (74%) | 0.956 |
| Smoking (Brinksman index>100) | 18 (51%) | 21 (45%) | 0.703 |
| Drinking Alcohol>14g/day * | 12 (34%) | 11(23%) | 0.403 |
| Medical history | | | |
| Hypertension | 20 (57%) | 29 (62%) | 0.85 |
| Diabetes mellitus | 6 (17%) | 16 (34%) | 0.145 |
| Hyperlipidemia | 13 (37%) | 25 (53%) | 0.223 |
| Dialysis | 3 (9%) | 1 (2%) | 0.411 |
| Cerebrovascular disease | 5 (14%) | 4 (9%) | 0.638 |
| Ischemic heart disease | 5 (14%) | 10 (21%) | 0.602 |
| Comorbidity index | 5 (0–9) | 5 (0–9) | 0.996 |
| Past history of Diverticulum bleeding | 18 (51%) | 18 (38%) | 0.337 |
| Anticoagulant drug | 6 (17%) | 3 (6%) | 0.236 |
| Antiplatelet drug | 7 (20%) | 13 (28%) | 0.59 |
| Number of bloody stools before hospitalization | 3 (1–10) | 4 (1–15) | 0.002 |
| Clinical examination | | | |
| White blood cell (10^3/μL) | 7.4 (4.7–15.7) | 7.4 (1.0–15.0) | 0.743 |
| Hemoglobin (g/dL) | 12.8 (5.6–15.9) | 10.9 (6.0–14.8) | 0.052 |
| Platelet (×10^4/uL) | 24.6 (12.4–46.8) | 19.8 (7.8–36.2) | 0.060 |
| Albumin (g/dL) | 3.9 (2.6–4.8) | 3.6 (2.3–4.7) | 0.089 |
| C-reactive protein (mg/dL) | 0.1 (0.0–2.8) | 0.1 (0.0–0.8) | 0.217 |
| Creatinine (mg/dL) | 0.8 (0.5–13.3) | 0.9 (0.4–8.2) | 0.268 |
| PT-INR | 1.0 (0.9–1.6) | 1.0 (0.9–1.9) | 0.368 |
| Physical examination | | | |
| Systolic blood pressure (mmHg) | 135 (89–181) | 121 (88–204) | 0.174 |
| Diastolic blood pressure (mmHg) | 80 (40–129) | 76 (47–117) | 0.072 |
| Heart rate (bpm) | 88 (53–120) | 82 (42–117) | 0.247 |
| Shock index | 0.65 (0.38–1.02) | 0.67 (0.21–1.07) | 0.978 |
| Shock index>1 | 1 (3%) | 2 (4%) | 1 |
| Contrast CT image of extravasation | 32 (91%) | 0 | <0.001 |

*Alcohol >14 g/day

≒Beer 350 mL

Data are expressed as number (percentage) or median (min–max).

p-values <0.05 were considered significant

addition to rebleeding and required 8 units of blood transfusion. The incidence of recurrent colonic diverticular bleeding within one month was 26% in the JD ≥3 group and 28% in the JD <3 group, with no significant difference (p = 1). The incidence of rebleeding after one month from discharge was higher in the JD ≥3 group (23% vs. 4% in the JD <3 group; p = 0.028). The incidence of adverse events other than rebleeding and the length of hospital stay were not significantly different. Only one adverse event was observed in the JD ≥3 group; the patient had urticaria after contrast agent use, which was mild and resolved without treatment. There were no other serious complications, or deaths.

**Table 2. Clinical outcomes between the JD ≥3 and JD <3 groups.**

| | JD ≥3 group (n = 35) | JD <3 group (n = 47) | p-value |
|---|---|---|---|
| Scoring Points | 3 (3–6) | 0 (0–2) | <0.001 |
| Number of patients with identified bleeding source | 27 (77%) | 11 (23%) | <0.001 |
| Bleeding site; C/A/T/D/S/R * | 2/15/2/1/7/0 | 0/9/0/0/2/0 | |
| Endoscopic hemostasis method; clipping method | 27 | 11 | 1 |
| other | 0 | 0 | 1 |
| Success rate of initial endoscopic hemostasis | 96% (26/27) | 100% (11/11) | 1 |
| Time from hospital visit to examination (hr) | 2.2 (1.2–8.0) | 18.2 (1.8–77.0) | <0.001 |
| Total endoscopic procedure time (min) | 70 (25–169) | 45 (19–112) | <0.001 |
| Bowel preparation | 2 (5%) | 40 (85%) | <0.001 |
| Number of patients taking oral anticoagulants | 6 (17%) | 3 (6%) | 0.236 |
| Anticoagulant reversal | 0 (0%) | 1 (33%) | 0.708 |
| Number of patients with blood transfusion | 11 (31%) | 19 (39%) | 0.545 |
| Amount of blood transfused (unit) | 0 (0–36) | 0 (0–12) | 0.568 |
| Number of patients who received ≥10 units of blood transfusions | 4 (11%) | 1 (2%) | 0.203 |
| Re-bleeding within 1 month | 9 (26%) | 13 (28%) | 1 |
| Re-bleeding after 1 month | 8 (23%) | 2 (4%) | 0.028 |
| Number of patients who underwent IVR during hospitalization | 5 (14%) | 0 | 0.027 |
| Number of patients admitted to the ICU | 1 (3%) | 0 | 0.882 |
| Number of hospitalization days | 6 (4–15) | 7 (3–17) | 0.579 |

*Location: Cecum/Ascending/Transverse/Descending/Sigmoid/Rectum

Data are expressed as number (percentage) or median (min–max)

p-values <0.05 were considered significant

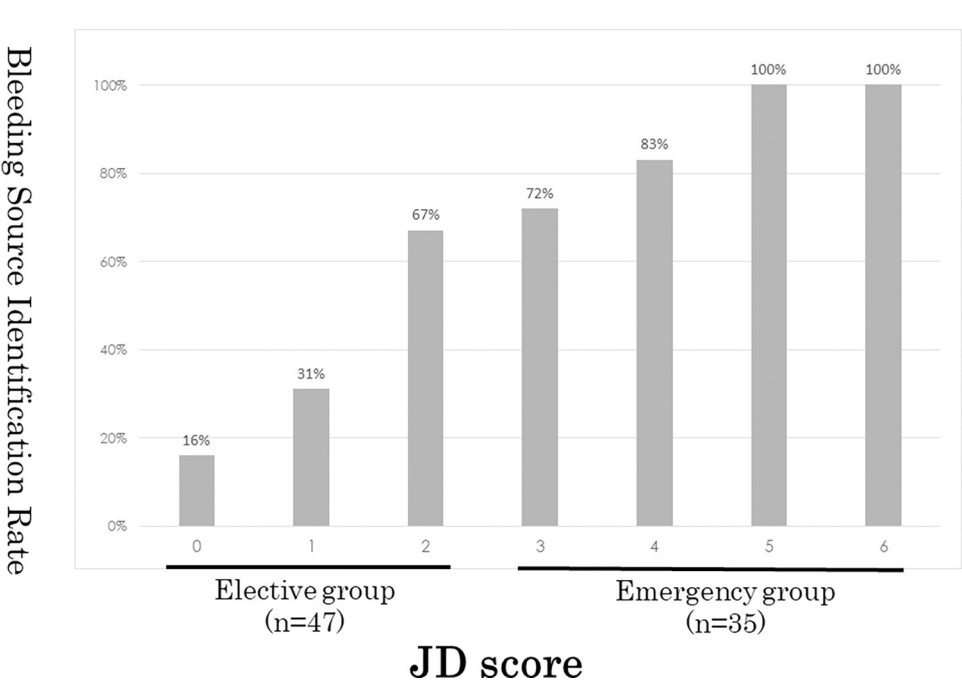

**Fig 2. Bleeding source identification rate by JD score points.**

## Subgroup analysis (identified group vs unidentified group)

Next, we performed a subgroup analysis to evaluate the clinical benefits of identifying the bleeding source to obtain hemostasis (Table 3). This analysis included 87 patients who underwent emergency or elective endoscopy for acute diverticular hemorrhage (JD ≥3 group, n = 35 patients; JD <3 group, n = 47 patients; and five patients who underwent endoscopy outside of the indicated study protocol period). The identified group comprised 41 patients with an identified source of bleeding. The unidentified group consisted of 46 patients with an unidentified source of bleeding. The identified group included many patients in the JD ≥3 group and was generally similar to the JD ≥3 group in terms of patient background, while the unidentified group included many patients in the JD <3 group. Patients in the identified group had higher JD scores and underwent endoscopy early after they arrived at the hospital without any pretreatment. All patients in the identified group underwent endoscopic hemostasis, and the success rate was as high as 98%. They also had significantly shorter hospital stays, 6 (3–15) vs. 7 (4–17) days, than the unidentified group (p = 0.018).

## Discussion

We performed a prospective evaluation of JD scores in the clinical management of patients with emergency or elective endoscopy and colonic diverticular hemorrhage. Compared to the JD <3 group, the patients in the JD ≥3 group had a higher JD score, which was associated with a higher bleeding source identification rate. The number of patients who required IVR and the incidence of rebleeding at one-month post-discharge were higher in the JD ≥3 group than in the JD <3 group. The subgroup analysis showed a shorter hospital stay for patients whose bleeding source was identified and for whom hemostatic treatment was performed.

### The bleeding source identification rate

In the JD ≥3 group, the bleeding source identification rate at the initial endoscopy was significantly higher than that in the JD <3 group, with higher scores associated with higher identification rates. This hemorrhage source identification rate (77%) was similar to that reported in our previous study (67%) and much higher than that reported in other studies, which ranges from 22−40% [24]. Considering that the overall bleeding source identification rate, including both the JD ≥3 and JD <3 groups, was 46% (38/82), which is comparable to other reports, we were able to efficiently select patients in whom the source of bleeding was likely to be identified. Subsequent hemostatic procedures were performed in all patients with the source of bleeding identified, and endoscopic hemostasis was successfully achieved in all cases but one. In one case of failure, IVR was implemented early, and hemostasis was achieved.

### The JD score and its components

The major factor leading to the classification in the JD ≥3 group was the presence of CT evidence of intestinal leakage of the contrast agent, which was positive in 91% of the patients in the JD ≥3 group. In the JD ≥3 group, only three patients were negative for CT evidence of intestinal leakage of the contrast agent. All patients had a score of 3 and were receiving oral anticoagulants (2 points), in addition to CRP ≥1 mg/dL in one patient and a comorbidity index ≥6 in two patients. Among these three patients, the bleeding source could be identified in only one patient. Although none of the JD score components, except for CT evidence of intestinal leakage, showed a significant association with the success of bleeding source identification, the bleeding source identification rate increased in proportion to the JD score.

**Table 3. Comparison of identified and non-identified groups of bleeding sources.**

| | Identified | Unidentified | p-value |
| --- | --- | --- | --- |
| | (n = 41) | (n = 46) | |
| **Patient Characteristics** | | | |
| Age | 74 (22–94) | 74 (40–91) | 0.515 |
| Male number | 28 (68%) | 35 (76%) | 0.568 |
| Smoking (Brinksman index>100) | 19 (46%) | 22 (48%) | 1 |
| Drinking Alcohol>14 g/day * | 10 (24%) | 14 (30%) | 0.697 |
| Medical history | | | |
| Hypertension | 23 (56%) | 30 (65%) | 0.516 |
| Diabetes mellitus | 11 (27%) | 13 (28%) | 1 |
| Hyperlipidemia | 16 (39%) | 24 (52%) | 0.311 |
| Dialysis | 4 (10%) | 0 | 0.098 |
| Cerebrovascular disease | 5 (12%) | 4 (9%) | 0.855 |
| Ischemic heart disease | 7 (17%) | 8 (17%) | 1 |
| Comorbidity index | 5 (0–9) | 5 (0–9) | 0.31 |
| Anticoagulant drug | 6 (15%) | 4 (9%) | 0.596 |
| Antiplatelet drug | 8 (20%) | 12 (26%) | 0.637 |
| Number of bloody stools before hospitalization | 3 (1–15) | 4 (1–10) | 0.013 |
| Clinical examination | | | |
| White blood cell (10^3/μL) | 7.6 (3.9–13.6) | 7.4 (1.0–15.7) | 0.465 |
| Hemoglobin (g/dL) | 10.9 (5.6–15.3) | 11.5 (6.0–15.9) | 0.822 |
| Platelet (10^4/uL) | 23.6 (9.3–41.3) | 22.1 (7.8–46.8) | 0.878 |
| Albumin (g/dL) | 3.7 (2.7–4.6) | 3.7 (2.3–4.8) | 0.768 |
| C-reactive protein (mg/dL) | 0.11 (0.02–2.81) | 0.10 (0.01–2.14) | 0.261 |
| Creatinine (mg/dL) | 0.8 (0.5–13.3) | 0.8 (0.4–1.3) | 0.832 |
| PT-INR | 1.0 (0.9–1.9) | 1.0 (0.9–2.3) | 0.526 |
| Physical examination | | | |
| Systolic blood pressure (mmHg) | 133 (94–182) | 128 (88–204) | 0.677 |
| Diastolic blood pressure (mmHg) | 78 (40–120) | 78 (47–129) | 0.792 |
| Heart rate (bpm) | 86 (53–120) | 82 (42–114) | 0.763 |
| Contrast CT image of extravasation | 26 (63%) | 6 (13%) | <0.001 |
| **Clinical outcomes** | | | |
| Scoring Points | 3 (0–6) | 0 (0–4) | <0.001 |
| Time from hospital visit to examination (hr) | 2.6 (1.2–48.2) | 16.1 (0.5–76.9) | <0.001 |
| Total Endoscopic procedure time (min) | 62 (25–169) | 45 (19–141) | 0.008 |
| Bowel preparation | 13 (32%) | 31 (67%) | 0.002 |
| Anticoagulant reversal | 0 | 1 (25%) | 0.83 |
| Endoscopic treatment | 41 (100%) | 0 | <0.001 |
| Success rate of endoscopic treatment | 40 (98%) | 0 | <0.001 |
| Number of patients with blood transfusion | 16 (39%) | 18(39%) | 1 |
| Amount of blood transfused (unit) | 0 (0–22) | 0 (0–36) | 0.782 |
| Number of patients who received ≥10 units of blood transfusions | 3 (7%) | 2 (4%) | 0.895 |
| Re-bleeding within 1 month | 10 (24%) | 14 (30%) | 0.633 |
| Re-bleeding after 1 month | 6 (15%) | 6 (13%) | 1 |
| Number of patients who underwent IVR during hospitalization | 4 (10%) | 1 (2%) | 0.291 |
| Number of patients admitted to the ICU | 0 | 1 (2%) | 1 |

(*Continued*)

**Table 3.** (Continued)

|  | Identified | Unidentified | p-value |
|---|---|---|---|
|  | (n = 41) | (n = 46) |  |
| Number of hospitalization days | 6 (3–15) | 7 (4–17) | 0.018 |

\* Alcohol >14g/day≒Beer 350mL

Data are expressed as number (percentage) or median (min−max).

p-values <0.05 were considered significant

CE-CT is an important factor of the JD score. Although there are some arguments concerning the clinical significance of CE-CT in patients with lower GI bleeding [39], its usefulness has been reported in several studies [40–42]. In this study, the examination was performed safely, without serious adverse events. CE-CT avoided unnecessary endoscopy in some patients through identifying other diseases, such as ischemic enteritis, and these patients were excluded from this study. Therefore, we consider that CE-CT is safe and useful if no contraindications exist. Patients intolerant to contrast agents were not included in this study, which is a limitation of our study.

## Emergency/elective endoscopy with the JD score and clinical outcomes

No significant differences were observed between the JD ≥3 and JD <3 groups in terms of clinical outcomes, except for a higher incidence of rebleeding at one-month post-discharge and a higher number of patients who required IVR in the JD ≥3 group than in the JD <3 group.

Our findings indicated that the longer time to endoscopy in the JD <3 group did not result in any clinical disadvantage when vital signs were stable, suggesting that the timing of elective endoscopy was not delayed. Emergency endoscopy did not provide any benefit in terms of clinical outcomes for the JD ≥3 group. Background patient characteristics did not differ between the two groups. However, more patients in the JD ≥3 group than in the JD <3 group requiring hemostasis with IVR might indicate that more potentially ill patients (in terms of disease factors) were included in the JD ≥3 group than in JD <3 group. Most patients in the JD ≥3 group showed extravasation on CE-CT scans, and this may have been related to more severe disease factors, such as ruptured vessels and fragility of the vessel wall. While emergency endoscopy followed by hemostasis might prevent further deterioration in a patient's condition, ectopic rebleeding may easily occur. In the JD ≥3 group, rebleeding was found to be more common at one-month post-discharge. No patients had a rebleeding event from the same diverticulum as that on initial bleeding, indicating that rebleeding originated from a different diverticulum site. The JD score may also be a predictor of the risk of rebleeding. This could be a useful tool to help alert patients to the risk of rebleeding during follow-up.

In a subgroup analysis, hospitalization was significantly shorter in the group in which the source of bleeding was identified and hemostatic procedures were performed. Therefore, if hemostatic treatment can prevent serious consequences and shorten hospitalization, it is worthwhile to use this scoring system to increase the pre-examination probability of bleeding source identification for efficient emergency endoscopy.

Although there have been several reports suggesting that emergency endoscopy is not useful in patients with lower GI bleeding [14, 21–23], these studies were conducted under different conditions from those in the present study, such as the inclusion of lower GI bleeding in a broad sense (not limited to diverticular bleeding), lower bleeding source identification rates than those in the present study, and emergency endoscopy was defined as being performed within 24 hours after presentation [14, 21, 25, 26, 43]. In this study, we sought to perform

emergency endoscopy earlier, as early endoscopy, performed within 5 h of presentation, was found to be associated with a higher bleeding source identification rate in our previous study [34]. Therefore, "within 5 hours" was a goal for implementation; however, this goal was not always strictly enforced to avoid an excessive burden on patients and medical personnel.

The JD $\geq$3 group had a longer endoscopic procedure time than the JD <3 group. It is possible that the procedure time was affected by bowel preparation. Compared with other studies [21, 44, 45], the procedure time in our study was slightly longer. This longer procedure time may have influenced the improvement in bleeding source identification rate, however, there are no reports comparing the procedure time to bleeding source identification rates; thus, no conclusions can be drawn.

The usefulness of endoscopic hemostatic procedures has been highlighted [13, 32], and considering the high bleeding source identification and hemostatic procedure rates, the effectiveness of the JD score cannot be denied. In this study, we were unable to compare outcomes with hemostatic methods other than endoscopic hemostasis, such as IVR and surgery. IVR is an efficient method for hemostasis, but the skill of the physician performing the IVR matters. Intestinal ischemia is reported to occur in 0–10% of cases as a complication, and other relatively serious adverse events have also been reported [46–49]. Therefore, IVR should be indicated only for patients who cannot be treated endoscopically or have rapidly progressing bleeding [9]. In this study, hemostasis was not achieved in one patient who underwent additional IVR after the initial endoscopy, and four patients underwent IVR for repeated rebleeding, but no patient was in such a poor general condition that endoscopy could not be performed at all. It is clinically meaningful to consider whether emergency endoscopy should be performed for suspected diverticular hemorrhage, which is the first-line treatment. Using the JD score to increase the pre-test probability of identifying the source of bleeding is helpful. When the JD score is low and vital signs are stable, endoscopy can be performed electively with no clinical disadvantages.

## Limitations

The limitations of this study include a small sample size, although the sample size was based on a pre-study analysis. CE-CT is a major component of the JD score, and this scoring is not useful in patients who are intolerant to contrast agents owing to conditions such as renal dysfunction and allergies. Also, the question of which patient need CE-CT remain open. A small number of extravascular leakage images of the contrast agent may be overlooked by emergency physicians. In addition, many endoscopists performed the endoscopic procedures, and their skills were not uniform, which may have affected the identification of the bleeding source. Most hemostatic procedures in this study involved clipping. Reports have shown that hemostasis using EBL is associated with a lower risk of rebleeding [50, 51]. The use of different endoscopic hemostatic techniques may have affected our study patients' clinical course. Further validation studies are needed to determine the efficacy of this clinical management system using JD scores in patients with acute colonic diverticular hemorrhage. We determined the cause of bleeding as diverticular hemorrhage only after having performed an endoscopic evaluation. In a real clinical setting, it is important to determine whether an emergency endoscopy should be performed for acute lower GI bleeding. Therefore, this study should be extended to the field of acute lower GI bleeding and cases without CE-CT.

## Conclusion

The introduction of the JD score led to a higher rate of bleeding source identification and hemostatic procedures in acute diverticular hemorrhage. Successful identification of the

bleeding source and hemostasis contributed to a short hospital stay. Selecting patients for whom there is a high possibility of identifying the bleeding source (JD ≥3) is helpful in clinical practice. In patients with a JD score <3, emergency endoscopy can be avoided without any clinical disadvantage when vital signs are stable, thereby reducing the burden on both patients and healthcare providers. The JD score is useful because it allows physicians (including non-specialists) to consider the need for emergency endoscopy to manage acute diverticular hemorrhage safely and efficiently in real-world settings.

## Supporting information

**S1 Checklist. TREND statement checklist.**
(DOC)

## Author Contributions

**Conceptualization:** Takeshi Uehara.

**Data curation:** Takeshi Uehara, Satohiro Matsumoto, Hiroyuki Tamura, Masahiro Kashiura, Takashi Moriya, Kenichi Yamanaka, Hakuei Shinhata, Masanari Sekine, Hiroyuki Miyatani.

**Formal analysis:** Takeshi Uehara.

**Supervision:** Hirosato Mashima.

**Writing – original draft:** Takeshi Uehara.

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
