## [Decision Letter · Decision Letter 0]

4 Dec 2022

PONE-D-22-25709A multicenter prospective study on the utility of a new scoring system to determine the adaptation of emergency endoscopy for colonic diverticular bleedingPLOS ONE

Dear Dr. UEHARA,

Thank you for submitting your manuscript to PLOS ONE. After careful consideration, we feel that it has merit but does not fully meet PLOS ONE’s publication criteria as it currently stands. Therefore, we invite you to submit a revised version of the manuscript that addresses the points raised during the review process.

ACADEMIC EDITOR:

The paper was found interesting in its original idea, but still has too many major concerns. Following the reviewers' suggestions, first it is necessary to better revise and structure the article with regard to scientific methodology. The M&M session requires more structural and scientifically presentation; the **statistical part** also needs to be better presented, scientifically, schematically and according to international standards. Consequently, the results must also follow the same logical presentation as in the M&M session. On discussion and conclusions, it is necessary to remain more factual. Last but not least, the introductory part needs to be better argued and make sure that each sentence has a background of scientific evidence without any speculation. Finally, reviewer #1's reflection on the background seems interesting to me; I think it needs to be explored further; if necessary, you can discuss it in the Discussion session.

As far as my critical thinking on the text is concerned, I prefer to expose my possible doubts to the Authors once the paper has been adjusted and revised according to this first absolutely necessary structural and content revision. As a leading journal, it is necessary not only to present an excellent paper, but to add clinical reflections and speculations that 'make a difference'. In order to do this, however, the structural and presentation basis must be as flawless as possible. At the moment, I ask the Authors to remain factual and resolve the doubts raised by the Reviewers; afterwards, we will consider whether a high-level speculative discussion is possible.

We look forward to receiving your revised manuscript.

Kind regards,

Samuele Ceruti

Academic Editor

PLOS ONE

Journal Requirements:

No

No 

4. Please ensure that you include a title page within your main document. You should list all authors and all affiliations as per our author instructions and clearly indicate the corresponding author.

5. Please upload a copy of your trial study protocol as a supporting information file. By the study protocol, we mean the complete and detailed plan for the conduct and analysis of the trial that the ethics committee approved before the trial began. Please send this in the original language. If this is in a language other than English, please also provide a translation. Please detail any deviations from this study protocol in the Methods section of your manuscript. Your study protocol will be made available to the editors and reviewers, and will be published as supporting information with your manuscript if accepted for publication. (If you do not agree to this, we will not be able to publish your manuscript). If you have formally published a study protocol for your trial in a journal then you should cite this in your manuscript, but you still need to send us the original document.

Reviewers' comments:

Reviewer's Responses to Questions

**Comments to the Author**

1. Is the manuscript technically sound, and do the data support the conclusions?

Reviewer #1: Partly

Reviewer #2: Yes

2. Has the statistical analysis been performed appropriately and rigorously? 

Reviewer #1: No

Reviewer #2: Yes

3. Have the authors made all data underlying the findings in their manuscript fully available?

Reviewer #1: Yes

Reviewer #2: Yes

4. Is the manuscript presented in an intelligible fashion and written in standard English?

Reviewer #1: Yes

Reviewer #2: Yes

5. Review Comments to the Author

Reviewer #1: Thank you to to the authors to submit this article. The authors evaluated the usefulness of a previously established score (the JD score) to detect the endoscopic source of bleeding in colonic diverticular bleeding. Unfortunately it is not very clear what was the aim of this research and accordingly, the design they used to state their hypothesis. E.g. in the abstract the authors stated that the aim of this study was "to establish an efficient and safe emergency endoscopy system", what it is neither clear, nor a scientific question. In the introduction they stated that the aim of this study was "to validate our scoring system" (i.e. JD score). The sentence before, the authors stated that "no study to date has examined whether emergency endoscopy should be performed for colonic diverticular bleeding using a scoring system". So, the hypothesis should be something challenging the indication of an emergency endoscopy, but the primary outcome was the identification of the bleeding source, which alone would have no impact. Why not using e.g. endoscopic hemostasis instead? This would be much more interesting. I read in the discussion first, that every patient in the emergency endoscopy group received endoscopic treatment, but exactly numbers and proportions in the elective endoscopy group are not given (which of course should be listed in the results). E.g. the article could be something like "Need for endoscopic treatment in acute colonic diverticular bleeding: a multicenter validation of the JD score". It would be however questionable to change the design of an intervention trial. Further, the JD score is based on CT scan findings, which is however not recommended for every patient with suspected diverticular bleeding (e.g. https://gut.bmj.com/content/gutjnl/68/5/776.full.pdf). Finally, 91% of the patients in the "emergency group" had an extravasation in the CT scan, what would of course increase the probability to see something in the endoscopy (what is the importance of the others parameters in the JD score?). The question of which patients need a CT scan remain open, unfortunately limiting the significance of this study.

Other comments

- methods: "to evaluate the benefit of hemostatic treatment, the patients were divided into those in whom the bleeding source could be identified (identified group) and those in whom it could not (non-identified group)" -> this is not clear, because identification of the bleeding source does not necessarily implicate hemostatic treatment. Additionally, there is no information about hemostatic treatment.

- an important number of patients received blood transfusions; however in the demographics (i.e. at least at presentation) none of the patient was unstable or with low hemoglobin. The authors should give more information about hemodynamic stability or low hemoglobin values in the course. It would be also interesting to know which patients needed ICU care.

- the power analysis is not clear. The authors should give more information. Additionally, the authors calculated 60 patients, but recruited 91 (?)

- the authors should give a reference for the comorbidity index

- results without importance (e.g. minimally statistically differences in platelets count or diastolic pressure) should not be reported in the text (is instead OK in the table)

- it would be interesting to know if anticoagulant reversal was performed in these patients

- how did the authors assess re-bleeding? There was a follow-up? How was performed?

- the limitations section should be expanded

- statements in the discussion e.g. by platelets and blood pressure are overreaching; additionally the authors should not repeat the results but stick to a global interpretation before assessing for external validity

- in the discussion the authors should compare endoscopic treatment vs endovascular treatment (especially when an extravasation is seen on the CT scan)

Reviewer #2: I would like to thank the editor for allowing me to review this paper by the Japanese group who thought to create an effective and safe emergency endoscopy system in diverticular bleeding.

1. I advise authors, when submitting a manuscript to a journal to always include the continuous line number so that reviewers can make a timely and accurate review.

2. For manuscript review, I entered the count on the submitted manuscript without making any other changes.

References

3. I advise authors to set the style of references as required by the journal. Perhaps using a computer system reference management tool so that it is done properly. https://journals.plos.org/plosone/s/submission-guidelines#loc-references

Title

4. I advise authors to edit the title.

Since, after reading the manuscript completely I understood that you wanted to do a validation study regarding the previously developed JD score so it would be important that the title also highlight that this is a validation study.

Keywords

5. I advise the authors to double-check through the MESH batabase the keywords:

a. colonic diverticular bleeding does not exist, however, Diverticulum Colon; Bleeding; Diverticular Diseases is present

b. scoring system does not exist, there is score; scoring;

c. emergency endoscopy does not exist, there is however: endoscopy; Endoscopy, Digestive System; Endoscopy, Gastrointestinal;

Introduction

6. Line 37: I suggest to the authors, as they included reference to the Japanese Gastroenterological Associatio guideline they should also include that of the American Gastroenterological Association (Strate LL, Gralnek IM. ACG Clinical Guideline: Management of Patients With Acute Lower Gastrointestinal Bleeding. Am J Gastroenterol. 2016 Apr;111(4):459-74. doi: 10.1038/ajg.2016.41. Epub 2016 Mar 1. Erratum in: Am J Gastroenterol. 2016 May;111(5):755. PMID: 26925883; PMCID: PMC5099081.) I'm not absolutely sure if this is the very latest version but it seems to be.

7. Line 38-39 Is this statement an inference by the authors or is it supported by the low evidence in the guidelines? I suggest the authors specify this better, perhaps putting the guideline passage with what it states and the level of evidence.

8. Line 64. I suggest the authors move the written references or add more in correlation to the statement made about "and size of the exposed blood vessel, whether the source of bleeding is an artery or vein, and the location of the diverticulum (visibility)." If these conditions are always integrated into the previously cited references I recommend citing them again, otherwise it seems like a statement by the authors themselves, without a rationale.

9. Line 64-67 I advise the authors to include references about it here as well or modify the sentence so that it is understood to be a statement by the authors.

10. Line 75. The score states that 1 point is taken for the use of oral anticoagulants. It would be important to perhaps specify which types, whether any type of anticoagulant or a particular type.

Materials and methods

11. Line 96-97 Why do the authors in point 2 of the exclusion criteria include "one or more episodes of bloody stools in the previous 30 days" They should justify why this is so.

12. Also at line 98 item 4 of the exclusion criteria what do the authors mean by "could not undergo endoscopy at the time indicated in the score." In what way, why could patients not undergo endoscopy? It would be better to justify

13. I would advise the authors to restructure the materials and methods with subchapters so as to make the work easier and more readable, for example with subchapters such as: inclusion and exclusion criteria, a subchapter with the JD score, perhaps tightening the part in the introduction and expanding the part in the methods, not forgetting that there is already a manuscript about it, so it should not be repetitive but quickly explanatory. I hope I have explained myself. One last subchapter with the two groups of patients, elective and emergency endoscopy.

14. With related subgroup developments. This would make the work more linear and logical and easily understood perhaps even by those not fully in the profession.

15. Another piece of advice I would give to the authors is to explain whether the election group was being "prepped" with a purgative in order to better assess the bowel or was it conservative without using laxatives for bowel cleansing. Because in Table 2 it is understood that 85% of the elective patients were bowel-prepared but it is not stated in the methods.

Discussion

16. Line 209-279 The authors make a long digression of the results obtained but unfortunately list the results of the study again in a descriptive manner without going into the results and data obtained but especially not making a comparison with the present literature.

17. The discussion should be a deepening of the results by comparing them with the present, or even past, literature, comparing their own results with those present, perhaps criticizing or pointing out the differences. The fact that no score is present does not mean that one cannot compare the results obtained, perhaps just with primary endpoint and then the secondary ones.

18. How is endoscopy time in the literature? bleeding after urgent or elective endoscopy? Or the days of hospitalization? Do they change from the literature? Transfusions?

19. I recommend that the authors remerge in the discussion and elaborate on the various points, not re-describing the results but elaborating on them.

Tables

20. I suggest the authors create legends for the tables, each table a title and an explanatory legend, not meaning to describe the results but to specify acronyms or data presentation and whatnot.

6. PLOS authors have the option to publish the peer review history of their article (what does this mean?). If published, this will include your full peer review and any attached files.

Reviewer #1: No

Reviewer #2: No

---

## [Author Response · Author response to Decision Letter 0]

6 Mar 2023

Thank you very much for reviewing our manuscript entitled “Need for emergency endoscopy in acute colonic diverticular bleeding: a multicenter validation of the JD score.” (PONE-D-22-25709, The title was corrected). We are grateful for the editor’s and reviewers’ thoughtful and valuable comments. We have corrected and revised our manuscript intensively in accordance with them. Data presentation in tables was changed from mean ± SD to median (min-max). We changed the method of statistical analysis from Student’s t-test to Mann-Whitney U test, because a group of patients was not based on a normal distribution. We have also addressed the issues raised by the reviewers in a point-by-point fashion, as outlined below. Red text in the main manuscript indicates the revised portions. We believe that these changes have greatly improved the quality and significance of our study and this version is now suitable for publication in PLOS ONE.

Reviewer #1: 

Unfortunately it is not very clear what was the aim of this research and accordingly, the design they used to state their hypothesis.

 E.g. in the abstract the authors stated that the aim of this study was "to establish an efficient and safe emergency endoscopy system", what it is neither clear, nor a scientific question. In the introduction they stated that the aim of this study was "to validate our scoring system" (i.e. JD score). The sentence before, the authors stated that "no study to date has examined whether emergency endoscopy should be performed for colonic diverticular bleeding using a scoring system". So, the hypothesis should be something challenging the indication of an emergency endoscopy, but the primary outcome was the identification of the bleeding source, which alone would have no impact. Why not using e.g. endoscopic hemostasis instead? This would be much more interesting. I read in the discussion first, that every patient in the emergency endoscopy group received endoscopic treatment, but exactly numbers and proportions in the elective endoscopy group are not given (which of course should be listed in the results).

Answer: 　Thank you for your valuable and important comments.

The aim of this study is to validate the usefulness of JD score, which indicates whether an emergency or elective endoscopy should be performed for colonic diverticular bleeding. Hemostatic treatment has been shown to improve patient outcomes and it requires the identification of the bleeding source in diverticular hemorrhage. Therefore, "bleeding source identification" was set as a primary outcome in this study and it is almost the same as hemostatic treatment. (Page 10, Line164-167)　Regarding the hemostatic treatment in the elective group, the bleeding source was identified in 11/47 (23%) and the hemostatic success rate was 11/11 (100%). we described these in “Results” section (Page 14, Line 230-237) and in Table 2.

E.g. the article could be something like "Need for endoscopic treatment in acute colonic diverticular bleeding: a multicenter validation of the JD score". It would be however questionable to change the design of an intervention trial.

Answer: 　Thank you very much for pointing out an important issue. In accordance with your advice,　we corrected　the title.

 Further, the JD score is based on CT scan findings, which is however not recommended for every patient with suspected diverticular bleeding Finally, 91% of the patients in the "emergency group" had an extravasation in the CT scan, what would of course increase the probability to see something in the endoscopy (what is the importance of the others parameters in the JD score?). The question of which patients need a CT scan remain open, unfortunately limiting the significance of this study.

Answer: 　Contrast-enhanced CT (CE-CT) is an important component and requirement for the JD score. Although there are some arguments concerning the clinical significance of CE-CT in patients with lower gastrointestinal bleeding, its usefulness has been reported in several studies (ref #41, #42, #43 in the manuscript). So, if there are no contraindications, such as renal dysfunction or allergy to contrast agent, we believe it acceptable to perform CE-CT in almost all cases. CE-CT sometimes detect some other disease, leading to avoid emergency endoscopy. 

There were only three cases without an extravasation in CE-CT in the emergency group. The other parameters in the JD score may have low significance in this scoring system. However, as shown in Figure 2, higher JD score were commensurate with a higher bleeding source identification rate. Therefore, these parameters help increase the pre-test probability of identifying the bleeding source. We added these descriptions in Page 20, Line 312-319. As you indicate, the question of which patients need a CT scan remain open. We added this point in the limitation of the study　in Page 23, Line 368-373.

Other comments

- methods: "to evaluate the benefit of hemostatic treatment, the patients were divided into those in whom the bleeding source could be identified (identified group) and those in whom it could not (non-identified group)" -> this is not clear, because identification of the bleeding source does not necessarily implicate hemostatic treatment. Additionally, there is no information about hemostatic treatment.

Answer: 　All cases with an identifiable source of bleeding were treated with hemostasis in this study. We also described about hemostatic treatment. We added these descriptions in “Materials and Methods” (Page 9, Line 156-161; Page 10, Line 164-167). 

- an important number of patients received blood transfusions; however in the demographics (i.e. at least at presentation) none of the patient was unstable or with low hemoglobin. The authors should give more information about hemodynamic stability or low hemoglobin values in the course. It would be also interesting to know which patients needed ICU care.

Answer: 　In acute hemorrhage, Hb often does not drop much at the time of presentation. We added shock index to “Table 1” as an indicator of the patient's hemodynamic stability. However, the indicator does not always indicate the severity of the patient's illness and no patient required ICU admission in this study (Page 15, Line 254-255). This may be due to appropriate post-arrival management, hemostasis procedures, blood transfusions, etc.

- the power analysis is not clear. The authors should give more information. Additionally, the authors calculated 60 patients, but recruited 91 (?)

Answer: 　The minimum sample size for this study was calculated as 60 and we set the study period long enough to accumulate these patient numbers. During that period, we have collected as many cases as possible to assure a more accurate evaluation. We added this description in Page 11, Line 184-187.

- the authors should give a reference for the comorbidity index

Answer: 　We added the reference (ref. #33).

- results without importance (e.g. minimally statistically differences in platelets count or diastolic pressure) should not be reported in the text (is instead OK in the table)

Answer: 　We changed the method of statistical analysis from Student’s t-test to Mann-Whitney U test, because a group of patients was not based on a normal distribution. Then, statistical differences in platelets count and diastolic pressure disappeared. So, we deleted these descriptions in “Results” section. 

- it would be interesting to know if anticoagulant reversal was performed in these patients

Answer: 　Anticoagulant reversal was performed in only one of the three patients in the elective group and none in the emergency group. We added this description column to Table 2 and Table 3.

- how did the authors assess re-bleeding? There was a follow-up? How was performed?

Answer: 　Rebleeding was defined as fresh bloody stools after endoscopy, which confirmed hemostasis, or bloody stools with a decrease of more than 2 g/dL of Hb. We added this description in Page 9, Line 161-162.

- the limitations section should be expanded

Answer: 　We added the descriptions in the limitation section, concerning the contraindications and reading skills of CT scan, and endoscopist’s skill (Page 23, Line 368-373)

- statements in the discussion e.g. by platelets and blood pressure are overreaching; additionally the authors should not repeat the results but stick to a global interpretation before assessing for external validity

Answer: 　We removed duplicate content in “Results” and “Discussion” sections. We revised the “Discussion” section considerably. 

- in the discussion the authors should compare endoscopic treatment vs endovascular treatment (especially when an extravasation is seen on the CT scan)

Answer: 　When we identified the source of bleeding, we initially applied clipping in all cases, though hemostatic procedure was selected at the discretion of endoscopists. In only one case, hemostasis was not achieved by clipping, and IVR was successfully performed. Therefore, we could not compare endoscopic treatment and any other treatment in this study. Then, we discussed IVR in “Discussion” section (Page 22-23, Line 350-362).

Reviewer #2: 

1. I advise authors, when submitting a manuscript to a journal to always include the continuous line number so that reviewers can make a timely and accurate review.

2. For manuscript review, I entered the count on the submitted manuscript without making any other changes.

Answer: 　According to your advice, we added line numbers in the manuscript.

References

3. I advise authors to set the style of references as required by the journal. Perhaps using a computer system reference management tool so that it is done properly. https://journals.plos.org/plosone/s/submission-guidelines#loc-references

Answer: 　We used “EndoNote” to properly fix the reference style.

Title

4. I advise authors to edit the title.

Since, after reading the manuscript completely I understood that you wanted to do a validation study regarding the previously developed JD score so it would be important that the title also highlight that this is a validation study.

Answer: 　Thank you very much for your advice. We corrected the title.

Keywords

5. I advise the authors to double-check through the MESH batabase the keywords:

a. colonic diverticular bleeding does not exist, however, Diverticulum Colon; Bleeding; Diverticular Diseases is present

b. scoring system does not exist, there is score; scoring;

c. emergency endoscopy does not exist, there is however: endoscopy; Endoscopy, Digestive System; Endoscopy, Gastrointestinal;

Answer: 　According to your advice, we corrected the key words

Introduction

6. Line 37: I suggest to the authors, as they included reference to the Japanese Gastroenterological Associatio guideline they should also include that of the American Gastroenterological Association (Strate LL, Gralnek IM. ACG Clinical Guideline: Management of Patients With Acute Lower Gastrointestinal Bleeding. Am J Gastroenterol. 2016 Apr;111(4):459-74. doi: 10.1038/ajg.2016.41. Epub 2016 Mar 1. Erratum in: Am J Gastroenterol. 2016 May;111(5):755. PMID: 26925883; PMCID: PMC5099081.) I'm not absolutely sure if this is the very latest version but it seems to be.

Answer: 　We added the guideline of American Gastroenterological Association as a reference (ref. #10). We also added the guideline of the European Society of Gastrointestinal Endoscopy (ref. #9).

7. Line 38-39 Is this statement an inference by the authors or is it supported by the low evidence in the guidelines? I suggest the authors specify this better, perhaps putting the guideline passage with what it states and the level of evidence.

Answer: 　As you pointed out, this statement is supported by the low level of evidence in the guidelines. We added this description in Page 4, Line 58-59

8. Line 64. I suggest the authors move the written references or add more in correlation to the statement made about "and size of the exposed blood vessel, whether the source of bleeding is an artery or vein, and the location of the diverticulum (visibility)." If these conditions are always integrated into the previously cited references I recommend citing them again, otherwise it seems like a statement by the authors themselves, without a rationale.

Answer: 　Thank you very much for pointing out. We corrected the content to be based on the literature and added the reference numbers properly. (Page 5, Line 83-89)

9. Line 64-67 I advise the authors to include references about it here as well or modify the sentence so that it is understood to be a statement by the authors.

Answer: 　This was our statement. We think it inappropriate to describe our statement in “Introduction”. Then, we deleted the description and modified wording. (Page 5, Line 83-89)

10. Line 75. The score states that 1 point is taken for the use of oral anticoagulants. It would be important to perhaps specify which types, whether any type of anticoagulant or a particular type.

Answer: 　The types of oral anticoagulant, such as DOAC and warfarin, were not specified in this study. We added “any type” in the “Abstract” (Page 2, Line 29) and “Materials and Methods” (Page 7, Line 124).

Materials and methods

11. Line 96-97 Why do the authors in point 2 of the exclusion criteria include "one or more episodes of bloody stools in the previous 30 days" They should justify why this is so.

Answer: 　Bleeding within 30 days may indicate a rebleeding. We excluded rebleeding cases, because the effects of previous bleeding would remain, and it may affect the future rebleeding. Although this is our speculation, we defined this as one of the exclusion criteria for this study. In fact, no cases were excluded with this criterion. We added this description in Page 7, Line113-115.

12. Also at line 98 item 4 of the exclusion criteria what do the authors mean by "could not undergo endoscopy at the time indicated in the score." In what way, why could patients not undergo endoscopy? It would be better to justify

Answer: 　As shown in Figure 1, two patients in JD score ≥3 did not undergo emergency endoscopy and were treated conservatively at the physician's discretion. Three patients in JD score <3 underwent emergency endoscopy at the discretion of their physicians due to a large amount of blood in the stool or unstable vital signs. We described these situations in “Results” section (Page 12, Line 202-208).

13. I would advise the authors to restructure the materials and methods with subchapters so as to make the work easier and more readable, for example with subchapters such as: inclusion and exclusion criteria, a subchapter with the JD score, perhaps tightening the part in the introduction and expanding the part in the methods, not forgetting that there is already a manuscript about it, so it should not be repetitive but quickly explanatory. I hope I have explained myself. One last subchapter with the two groups of patients, elective and emergency endoscopy.

Answer: 　Thank you very much for pointing-out an important issue. According to your advice, we restructured “Materials and Methods” and “Results” sections with subchapters. We think these changes have made our study easier and more readable.

14. With related subgroup developments. This would make the work more linear and logical and easily understood perhaps even by those not fully in the profession.

Answer: 　With a subchapter of “Subgroup analysis” in “Results” section, we think it became easy to read (Page 16-17, Line 264-274). As mentioned before, we changed the method of statistical analysis (Page 11, Line188-189). Then, it is found that the identified group had a significantly shorter hospital stay than the unidentified group (Page 17, Line 273-274).

15. Another piece of advice I would give to the authors is to explain whether the election group was being "prepped" with a purgative in order to better assess the bowel or was it conservative without using laxatives for bowel cleansing. Because in Table 2 it is understood that 85% of the elective patients were bowel-prepared but it is not stated in the methods.

Answer: 　We explained about bowel preparation in the “Materials and Methods” (Page 8-9, Line 138-146) .

Discussion

16. Line 209-279 The authors make a long digression of the results obtained but unfortunately list the results of the study again in a descriptive manner without going into the results and data obtained but especially not making a comparison with the present literature.

17. The discussion should be a deepening of the results by comparing them with the present, or even past, literature, comparing their own results with those present, perhaps criticizing or pointing out the differences. The fact that no score is present does not mean that one cannot 

compare the results obtained, perhaps just with primary endpoint and then the secondary ones.

Answer: 　Thank you very much again for pointing-out an important issue. We corrected the entire ”Discussion” section with subchapters. We eliminated duplicates with "Results" section and compared our results with the present literature. 

18. How is endoscopy time in the literature? bleeding after urgent or elective endoscopy? Or the days of hospitalization? Do they change from the literature? Transfusions?

Answer: 　There was no coherent paper on endoscopic procedure time, and it was only mentioned briefly in some papers. The endoscopic procedure time tended to be longer in our study, compared to other studies. This may be due to our willingness to identify the source of bleeding. Although many papers have found no significant changes in clinical outcomes in emergency endoscopy, patient backgrounds and results differ significantly from paper to paper, making simple comparisons difficult. We added this description in Page 22, Line 342-347. 

19. I recommend that the authors remerge in the discussion and elaborate on the various points, not re-describing the results but elaborating on them.

Answer: 　As in the response in 16 and 17, we have revised the “Discussion” section in its entirety.

Tables

20. I suggest the authors create legends for the tables, each table a title and an explanatory legend, not meaning to describe the results but to specify acronyms or data presentation and whatnot.

Answer: 　Thank you very much for pointing out. We modified the tables and created legends, as you indicated.

---

## [Decision Letter · Decision Letter 1]

27 Mar 2023

PONE-D-22-25709R1Need for emergency endoscopy in acute colonic diverticular bleeding: a multicenter validation of the JD score.PLOS ONE

Dear Dr. UEHARA,

Thank you for submitting your manuscript to PLOS ONE. After careful consideration, we feel that it has merit but does not fully meet PLOS ONE’s publication criteria as it currently stands. Therefore, we invite you to submit a revised version of the manuscript that addresses the points raised during the review process. I thank the Authors for this interesting and potentially useful pilot study for the entire scientific community. The study is, however, burdened by an approximation in the methodological presentation that in my opinion needs to be clarified and resolved; the most important advice is to involve an additional expert colleague who can give a methodological revision in the presentation of the paper. Specifically: a) in M&M it is necessary to immediately clarify the Multicenter study; the two hospitals mentioned must be framed as dimensions (Primary Care, Tertiary Care, etc...) and it is necessary to know approximately how many endoscopists were involved (one per centre? more endoscopists per centre? etc...). b) the allocation is never clearly explained, which is only apparent from Figure 1. It must be clearly explained that based on the JD score, it is determined how to treat patients (elective vs urgent). This point is fundamental. c) There are patients who are excluded because they are then managed differently from what was initially planned. They must be reported in absolute value (as has been done), but also in percentage. This shows the limitations of the JD score, which will perhaps require further study in the future with additional parameters to avoid having these exclusions which are significant in percentage terms. d) What reviewer #2 pointed out is fundamental and should be placed as a clear limitation in the methodology.

e) The Authors should reflect on whether it would not be better to present this paper as a Pilot Study, even in the title. The numbers are objectively low and it would be better to call it that.

I find the paper interesting, but there is a lack of methodological clarity in the exposition. I would therefore ask the authors to tidy up the M&M session, put this part in order as if it were the first time we were re-reading this paper - explaining well each step in the: 1. selection of patients (inclusion/exclusion), 2. allocation of patients according to JD, 3. taking care of patients medically (crystalloids, transfusions, antibiotics, etc.) and 4. when you are further excluded. Because it is unclear to understand the pathway.

Consequently, the presentation of the results must also be ordered as in M&Ms, and the limits must be adjusted.

I mark the paper as a Major Revision because the work to be done is still structural, but I strongly encourage the Authors to adapt this revision and then proceed with the next steps.

We look forward to receiving your revised manuscript.

Kind regards,

Samuele Ceruti

Academic Editor

PLOS ONE

Reviewers' comments:

Reviewer's Responses to Questions

**Comments to the Author**

1. If the authors have adequately addressed your comments raised in a previous round of review and you feel that this manuscript is now acceptable for publication, you may indicate that here to bypass the “Comments to the Author” section, enter your conflict of interest statement in the “Confidential to Editor” section, and submit your "Accept" recommendation.

Reviewer #1: All comments have been addressed

Reviewer #2: All comments have been addressed

2. Is the manuscript technically sound, and do the data support the conclusions?

Reviewer #1: Partly

Reviewer #2: (No Response)

3. Has the statistical analysis been performed appropriately and rigorously? 

Reviewer #1: No

Reviewer #2: (No Response)

4. Have the authors made all data underlying the findings in their manuscript fully available?

Reviewer #1: Yes

Reviewer #2: Yes

5. Is the manuscript presented in an intelligible fashion and written in standard English?

Reviewer #1: Yes

Reviewer #2: Yes

6. Review Comments to the Author

Reviewer #1: Thank you to the authors to resubmit a revised version of the manuscript. Unfortunately there is a major design concern, which appears not possible to address.

The JD score was developed to predict the endoscopic identification of the bleeding source in patients with suspected colonic diverticular bleeding. A validation study should accordingly prospectively confirm that patients with a JD score ≥ 3 have an higher endoscopic identification rate of the bleeding source in patients with suspected colonic diverticular bleeding (compared with JD score < 3). This means that each patient should undergo endoscopy and categorized as JD score ≥ 3 (or JD score < 3), and identification of the bleeding source (yes/no). However, at least two patients (with JD score ≥ 3) did not undergo emergency endoscopy and were treated conservatively at the physician's discretion. A physician-based decision in a validation study is however not acceptable. Further, the authors divided patients in an emergency endoscopy vs elective endoscopy group. This seems to refer to another outcome than the identification of the bleeding source (e.g. identification of patients who need emergency endoscopy). If so (which again is not validation of the score), the authors should state another hypothesis and especially another power calculation. Additionally, each patient with a JD score ≥ 3 should undergo emergency endoscopy, and each patient with a JD score < 3 should not. Both of these conditions were not applied to this study.

Reviewer #2: Certainly an adequate level of review has been carried out by the authors.

The work certainly appears to be much more supported by scientific evidence and evidence also present in the literature, although weak, but stated by the authors with a less speculative development of the work and the results obtained by relying solely on the present results.

7. PLOS authors have the option to publish the peer review history of their article (what does this mean?). If published, this will include your full peer review and any attached files.

Reviewer #1: No

Reviewer #2: No

---

## [Author Response · Author response to Decision Letter 1]

8 May 2023

The following is described in the Responce letter.

Thank you very much for reviewing our manuscript entitled “Need for emergency endoscopy in acute colonic diverticular bleeding: a multicenter validation of the JD score.” (PONE-D-22-25709)

We are grateful for the editor’s and reviewers’ thoughtful and valuable comments. As pointed out, this was not a validation study. We should treat this study as a pilot study and have changed the title. We have corrected and revised our manuscript intensively in accordance with your comments. 

We believe that these changes have greatly improved the quality and significance of our study and this version is now suitable for publication in PLOS ONE.

Editor’s comments:

a) in M&M it is necessary to immediately clarify the Multicenter study; the two hospitals mentioned must be framed as dimensions (Primary Care, Tertiary Care, etc...) and it is necessary to know approximately how many endoscopists were involved (one per centre? more endoscopists per centre? etc...).

Answer: 　Thank you for your valuable comments. We immediately clarified in M&M that this is a prospective multicenter study (Page 6, Line 107). We have added the information about dimensions and the number of endoscopists per center in Page 6-7, Line 109-112.

b) the allocation is never clearly explained, which is only apparent from Figure 1. It must be clearly explained that based on the JD score, it is determined how to treat patients (elective vs urgent). This point is fundamental.

Answer: 　 Thank you very much for your valuable and important comments. In the section of “Allocation of patients according to the JD score” in M&M, we clearly explained the flow of patients and the way of grouping according to the JD score (Page 8, Line 131-146). We have also modified Figure 1 to better reflect the actual clinical flow.

c) There are patients who are excluded because they are then managed differently from what was initially planned. They must be reported in absolute value (as has been done), but also in percentage. This shows the limitations of the JD score, which will perhaps require further study in the future with additional parameters to avoid having these exclusions which are significant in percentage terms.

Answer: 　Thank you very much for your important comments. We have added the percentage of patients excluded in addition to the absolute value (Page 13, Line 223-225). Since the JD score was designed for diverticular hemorrhage, we excluded cases other than diverticular hemorrhage in order to ascertain the effect of the therapeutic intervention on the score (Page 8, Line 143-144). However, it was unclear in real clinical practice whether lower GI bleeding was due to diverticular bleeding before performing lower GI endoscopy. Therefore, it is important to know whether emergency endoscopy should be performed in patients with lower GI bleeding. It is worthwhile to evaluate the usefulness of JD score with other parameters in patients with lower GI bleeding in the future. Also, we need to extend our study to patients without CE-CT and to evaluate the need of CE-CT in clinical settings. Thank you very much for showing us the future directions.

d) What reviewer #2 pointed out is fundamental and should be placed as a clear limitation in the methodology

Answer: 　Thank you very much for your valuable comment. As you noted, this study has many limitations. We added several limitations to show a clear limitation in the methodology（Page 24　Line 384-393）

e) The Authors should reflect on whether it would not be better to present this paper as a Pilot Study, even in the title. The numbers are objectively low and it would be better to call it that.

Answer: 

As you and reviewer #1 pointed out, this study did not meet validation studies. This study should be treated as a pilot study. So, we changed our title to “Evaluation of the Jichi Medical University diverticular hemorrhage score in the clinical management of acute diverticular bleeding using endoscopy: a pilot study.”

I find the paper interesting, but there is a lack of methodological clarity in the exposition. I would therefore ask the authors to tidy up the M&M session, put this part in order as if it were the first time we were re-reading this paper - explaining well each step in the: 1. selection of patients (inclusion/exclusion), 2. allocation of patients according to JD, 3. taking care of patients medically (crystalloids, transfusions, antibiotics, etc.) and 4. when you are further excluded. Because it is unclear to understand the pathway. Consequently, the presentation of the results must also be ordered as in M&Ms, and the limits must be adjusted.

Answer: Thank you very much for your important comments. As you indicated, we have restructured the M&M session. We have created several paragraphs to make the text more readable and understandable. We also revised the Result, Discussion, Limitations, and Conclusion sections.

6. Review Comments to the Author

Reviewer #1: Thank you to the authors to resubmit a revised version of the manuscript. Unfortunately there is a major design concern, which appears not possible to address. The JD score was developed to predict the endoscopic identification of the bleeding source in patients with suspected colonic diverticular bleeding. A validation study should accordingly prospectively confirm that patients with a JD score ≥ 3 have an higher endoscopic identification rate of the bleeding source in patients with suspected colonic diverticular bleeding (compared with JD score < 3). This means that each patient should undergo endoscopy and categorized as JD score ≥ 3 (or JD score < 3), and identification of the bleeding source (yes/no). However, at least two patients (with JD score ≥ 3) did not undergo emergency endoscopy and were treated conservatively at the physician's discretion. A physician-based decision in a validation study is however not acceptable. Further, the authors divided patients in an emergency endoscopy vs elective endoscopy group. This seems to refer to another outcome than the identification of the bleeding source (e.g. identification of patients who need emergency endoscopy). If so (which again is not validation of the score), the authors should state another hypothesis and especially another power calculation. Additionally, each patient with a JD score ≥ 3 should undergo emergency endoscopy, and each patient with a JD score < 3 should not. Both of these conditions were not applied to this study.

Answer: 　Thank you very much for your valuable and important comments. As you indicated, this study did not meet validation studies. This study should be treated as a pilot study. We corrected the title and revised the main text intensively.

Since the JD score was designed for diverticular hemorrhage, we excluded cases other than diverticular hemorrhage in order to ascertain the effect of the therapeutic intervention on the score (Page 8, Line 136-146). The main objective of this trial is to reduce the number of emergency endoscopies without disadvantage to patients and to reduce the burden on patients and medical staff. However, since these items are difficult to quantify, we hypothesized that patients with a high probability of identifying the bleeding source should undergo emergency endoscopy, and patients with a low probability do not need to undergo emergency endoscopy, but elective endoscopy. Therefore, the bleeding source identification rate, leading to the hemostatic procedure, was set as the primary endpoint (Page10; Line 171-178).

We performed power calculation based on the bleeding source identification rate in our previous study. We apologize for the error in the power calculation. The sample size was recalculated, and the sample size required to detect a significant difference in the emergency endoscopy was 27, not 15 (Page 11; Line190-198). In this study, patients with a JD score <3 were assigned to receive elective endoscopy. With the extrapolation of the result of the previous study, the total required sample size was calculated to be 81. We corrected the number and revised the descriptions in “Statistical analysis” of M&M section.

Reviewer #2: Certainly an adequate level of review has been carried out by the authors.

The work certainly appears to be much more supported by scientific evidence and evidence also present in the literature, although weak, but stated by the authors with a less speculative development of the work and the results obtained by relying solely on the present results.

Answer: 　Thank you very much for your valuable and important comments. As you pointed out, this study did not meet validation studies. This study should be treated as a pilot study. We corrected the title and revised the main manuscript intensively. We revised the “Limitations” section to show a clear limitation in the methodology. Further studies are needed to validate out results and this study should be extended to the field of acute lower GI bleeding and cases without CE-CT. We also need to evaluate the indication of CE-CT in real clinical settings.

---

## [Decision Letter · Decision Letter 2]

29 May 2023

PONE-D-22-25709R2Evaluation of the Jichi Medical University diverticular hemorrhage score in the clinical management of acute diverticular bleeding using endoscopy: a pilot studyPLOS ONE

Dear Dr. UEHARA,

Thank you for submitting your manuscript to PLOS ONE. After careful consideration, we feel that it has merit but does not fully meet PLOS ONE’s publication criteria as it currently stands. Therefore, we invite you to submit a revised version of the manuscript that addresses the points raised during the review process.

ACADEMIC EDITOR:I kindly ask the Authors to be extremely precise and clear in their answers to all reviewers, especially to Reviewer #1 in order to process the request. The delay in journal replying was mainly due to a strong disagreement between Reviewers. I think the paper should go ahead, but with the need to process the reviewer's requests and smooth out some of the edges of the text, according to Reviewer #1 suggestion.

We look forward to receiving your revised manuscript.

Kind regards,

Samuele Ceruti

Academic Editor

PLOS ONE

Journal Requirements:

Reviewers' comments:

Reviewer's Responses to Questions

**Comments to the Author**

1. If the authors have adequately addressed your comments raised in a previous round of review and you feel that this manuscript is now acceptable for publication, you may indicate that here to bypass the “Comments to the Author” section, enter your conflict of interest statement in the “Confidential to Editor” section, and submit your "Accept" recommendation.

Reviewer #1: All comments have been addressed

Reviewer #2: All comments have been addressed

Reviewer #3: (No Response)

2. Is the manuscript technically sound, and do the data support the conclusions?

Reviewer #1: Partly

Reviewer #2: Yes

Reviewer #3: Yes

3. Has the statistical analysis been performed appropriately and rigorously? 

Reviewer #1: No

Reviewer #2: Yes

Reviewer #3: Yes

4. Have the authors made all data underlying the findings in their manuscript fully available?

Reviewer #1: Yes

Reviewer #2: Yes

Reviewer #3: Yes

5. Is the manuscript presented in an intelligible fashion and written in standard English?

Reviewer #1: Yes

Reviewer #2: Yes

Reviewer #3: Yes

6. Review Comments to the Author

Reviewer #1: Thank you to the authors for this resubmission. This study still not show a clear hypothesis, which has been tested. Starting from the title, the «clinical management … using endoscopy» has for me no meaning. What I previously suggested, was the validity of the score to determine emergent endoscopy. However the score has not been developed for this. The authors showed in the course of this review a "data driven" approach, instead of an "hypothesis driven" approach, what would have been preferable. Accordingly, conclusion are not based on the data, but are derived indirectly. E.g. stating that "...patients with a JD score <3, emergency endoscopy can be avoided without any clinical disadvantage" as a conclusion is fault. In fact, 3 patients with a JD score < 3 needed emergency endoscopy. Further, the hospitalization was shorter for patients with identified bleeding source, BUT was not shorter based on the JD score (Table 2). Finally, what would have happened if patients with a JD score > 3 would have received elective endoscopy? In fact, 2 patients underwent elective endoscopy and in the group with a JD score < 3, 11 patients had an identified bleeding source necessitating clipping. This study accordingly does not determine the need for emergency lower GI endoscopy. There is still some data that are difficult to imagine, e.g. no patient had ICU care, but at least one patient received 36 (!) unit of blood (Table 2). Last but not least, the authors should be more precise on the studied population. E.g. the JD score was develop for diverticular bleeding, but the authors use "lower GI" interchangeable along the paper.

Reviewer #2: This further review of the work perhaps places it in its own specific domain. A pilot study for an initial investigation toward identifying a possible categorization of this type of patient. To accept the work as a multicenter study would have been too much. As a pilot study is the right placement of the work done.

Reviewer #3: The utility of the JD score devised for patients with diverticular bleeding was evaluated. The primary and secondary endpoints were the bleeding source identification rate and clinical outcomes. The rate of bleeding source identification was significantly higher in the JD ≥3 group than in the JD <3 group. The groups did not differ significantly with respect to clinical outcomes.

Minor revisions:

1- Line 191: State the statistical testing method which achieves 80% power.

2- Line 204: Use a lower case p for “p-values”.

3- For some comparisons of proportions, chi-square tests may be more powerful than Fisher’s exact tests, considering the expected sample sizes.

7. PLOS authors have the option to publish the peer review history of their article (what does this mean?). If published, this will include your full peer review and any attached files.

Reviewer #1: No

Reviewer #2: No

Reviewer #3: No

---

## [Author Response · Author response to Decision Letter 2]

11 Jul 2023

We have revised the paper according to your suggestion.

Although the content is poor, it is the result of my best efforts.

Please refer to the attached Letter.

Thank you very much for your cooperation.

---

## [Decision Letter · Decision Letter 3]

25 Jul 2023

Evaluation of the Jichi Medical University diverticular hemorrhage score in the clinical management of acute diverticular bleeding with emergency or elective endoscopy: a pilot study

PONE-D-22-25709R3

Dear Dr. UEHARA,

We’re pleased to inform you that your manuscript has been judged scientifically suitable for publication and will be formally accepted for publication once it meets all outstanding technical requirements.

Kind regards,

Samuele Ceruti

Academic Editor

PLOS ONE

Additional Editor Comments (optional):

Reviewers' comments:

Reviewer's Responses to Questions

**Comments to the Author**

1. If the authors have adequately addressed your comments raised in a previous round of review and you feel that this manuscript is now acceptable for publication, you may indicate that here to bypass the “Comments to the Author” section, enter your conflict of interest statement in the “Confidential to Editor” section, and submit your "Accept" recommendation.

Reviewer #2: All comments have been addressed

Reviewer #3: All comments have been addressed

2. Is the manuscript technically sound, and do the data support the conclusions?

Reviewer #2: Yes

Reviewer #3: (No Response)

3. Has the statistical analysis been performed appropriately and rigorously? 

Reviewer #2: Yes

Reviewer #3: (No Response)

4. Have the authors made all data underlying the findings in their manuscript fully available?

Reviewer #2: Yes

Reviewer #3: (No Response)

5. Is the manuscript presented in an intelligible fashion and written in standard English?

Reviewer #2: Yes

Reviewer #3: (No Response)

6. Review Comments to the Author

Reviewer #2: (No Response)

Reviewer #3: (No Response)

7. PLOS authors have the option to publish the peer review history of their article (what does this mean?). If published, this will include your full peer review and any attached files.

Reviewer #2: No

Reviewer #3: No

---

## [Editor Report · Acceptance letter]

14 Aug 2023

PONE-D-22-25709R3 

Evaluation of the Jichi Medical University diverticular hemorrhage score in the clinical management of acute diverticular bleeding with emergency or elective endoscopy: a pilot study 

Dear Dr. Uehara:

I'm pleased to inform you that your manuscript has been deemed suitable for publication in PLOS ONE. Congratulations! Your manuscript is now with our production department. 

Kind regards, 

on behalf of

Dr. Samuele Ceruti 

Academic Editor

PLOS ONE